# Oridonin is a covalent NLRP3 inhibitor with strong anti-inflammasome activity

Hongbin He[1,2], Hua Jiang[1], Yun Chen[3], Jin Ye[1], Aoli Wang[4], Chao Wang [1], Qingsong Liu [4], Gaolin Liang [5], Xianming Deng[3], Wei Jiang[1] & Rongbin Zhou [1,2]

Oridonin (Ori) is the major active ingredient of the traditional Chinese medicinal herb *Rabdosia rubescens* and has anti-inflammatory activity, but the target of Ori remains unknown. NLRP3 is a central component of NLRP3 inflammasome and has been involved in a wide variety of chronic inflammation-driven human diseases. Here, we show that Ori is a specific and covalent inhibitor for NLRP3 inflammasome. Ori forms a covalent bond with the cysteine 279 of NLRP3 in NACHT domain to block the interaction between NLRP3 and NEK7, thereby inhibiting NLRP3 inflammasome assembly and activation. Importantly, Ori has both preventive or therapeutic effects on mouse models of peritonitis, gouty arthritis and type 2 diabetes, via inhibition of NLRP3 activation. Our results thus identify NLRP3 as the direct target of Ori for mediating Ori's anti-inflammatory activity. Ori could serve as a lead for developing new therapeutics against NLRP3-driven diseases.

[1] Hefei National Laboratory for Physical Sciences at Microscale, the CAS Key Laboratory of Innate Immunity and Chronic Disease, School of Life Sciences, University of Science and Technology of China, Hefei 230027, China. [2] Innovation Center for Cell Signaling Network, University of Science and Technology of China, Hefei 230027, China. [3] State Key Laboratory of Cellular Stress Biology, Innovation Center for Cell Signaling Network, School of Life Sciences, Xiamen University, Xiamen, Fujian 361102, China. [4] High Magnetic Field Laboratory, Chinese Academy of Sciences, Hefei, Anhui 230031, China. [5] Hefei National Laboratory for Physical Sciences at Microscale, Department of Chemistry, University of Science and Technology of China, Hefei 230026, China. These authors contributed equally: Hongbin He, Hua Jiang, Yun Chen. These authors jointly supervised this work: Xianming Deng, Wei Jiang, Rongbin Zhou. Correspondence and requests for materials should be addressed to X.D. (email: xmdeng@xmu.edu.cn) or to W.J. (email: ustcjw@ustc.edu.cn) or to R.Z. (email: zrb1980@ustc.edu.cn)

O ridonin (Ori), a bioactive ent-kaurane diterpenoid, is the major active constituent of *Rabdosia rubescens*, which has been widely used in traditional Chinese medicine[1,2]. Ori has shown considerable anticancer activities, which include cell cycle arrest, apoptosis induction, and angiogenesis suppression, but the relatively moderate potency and imprecise mechanisms of action have greatly hindered its clinical applications for the treatment of cancer[3,4]. Besides their antitumor activity, *Rabdosia rubescens* and Ori have also possessed anti-inflammatory activity. In China, *Rabdosia rubescens* is a commonly available over-the-counter (OTC) herbal medicine for the treatment of inflammatory diseases[5,6]. Ori has been reported to inhibit NF-κB or MAPK activation to suppress the release of proinflammatory cytokines, such as tumor necrosis factor (TNF)-α and interleukin (IL)-6[7–9]. Moreover, Ori has also exhibited anti-inflammatory activity and protective role in colitis, sepsis, and neuroinflammation[10–13]. Although the anti-inflammatory activity of Ori is emerging, the underlying mechanisms and direct target are unknown.

NLRP3 inflammasome is a multiple protein complex composed of innate immune sensor NLRP3, ASC, and caspase-1[14–17]. The assembly of this complex results in the activation of caspase-1, which promotes the cleavage of pro-IL-1β and pro-IL-18 to produce mature and functional IL-1β and IL-18[18,19], so it plays a central role in innate immunity and inflammation. NLRP3 can be activated by many different stimuli including both danger-associated molecular patterns (DAMPs) and pathogen-associated molecular patterns (PAMPs)[14–16,18]. These stimuli seem to converge on three main processes intracellular ionic fluxes, ROS generation, and lysosomal damage to activate NLRP3 inflammasome, but the precise mechanism of NLRP3 activation is still unclear[14–16,18]. In addition, the aberrant NLRP3 inflammasome activation contributes to the progress of several human diseases, such as type 2 diabetes (T2D), atherosclerosis, Gout, and neurodegenerative diseases[20–26]. Moreover, several NLRP3 inflammasome inhibitors have been reported to exert beneficial effects for NLRP3-related diseases in animal models[27–33]. Some herbs used in traditional Chinese medicine have shown good anti-inflammatory activity and beneficial effects on inflammatory diseases[34], but the complex composition and unclear pharmacology limit their clinical application.

In this study, we show that Ori can target NLRP3 to exert its anti-inflammatory activity. On covalently binding to Cys279, Ori blocks the NLRP3–NEK7 interaction and the subsequent NLRP3 inflammasome assembly and activation, leading to an effective suppression of NLRP3-related diseases.

## Results

**Ori specifically inhibits NLRP3 inflammasome activation.** To test whether oridonin (Ori) could block NLRP3 activation, we examined the impact of Ori on caspase-1 activation and IL-1β secretion (Fig. 1a). Ori exhibited dose-dependent inhibitory effects on caspase-1 cleavage, IL-1β secretion, and cell death when treated with nigericin at the doses of 0.5–2 μM in lipopolysaccharides (LPS)-primed bone marrow–derived macrophages (BMDMs), but had no effects on inflammasome-independent cytokine TNF-α production (Fig. 1b–d and Supplementary Fig. 2A). Treatment with Ori also inhibited caspase-1 activation and IL-β release stimulated by other NLRP3 agonists, including monosodium urate crystals (MSU), ATP or cytosolic LPS (cLPS) (Fig. 1e, f and Supplementary Fig. 2B, C)[15,35], suggesting Ori as a broad-spectrum inhibitor of NLRP3 inflammasome. We also tested whether Ori was working for human cells. LPS-induced caspase-1 activation and IL-1β secretion in human peripheral blood mononuclear cells (PBMCs) could also be suppressed by

Ori (Fig. 1g, h). In contrast, Ori had no effects on TNF-α production (Fig. 1i). The same results were obtained in PBMCs from other three donors (Supplementary Fig. 1). We also found that Ori could not suppress NLRC4 or AIM2 inflammasome activation induced by *Salmonella typhimurium* (*Salmonella*) infection or poly A:T transfection, respectively (Supplementary Fig. 2D–G).

Previous studies have shown that Ori can inhibit NF-κB activation[7–9], we then examined whether Ori affected LPS-induced priming for inflammasome activation. When BMDMs were stimulated with Ori at the doses of 0.5–2 μM before or after LPS treatment, Ori could not suppress LPS-induced pro-IL-1β, NLRP3 expression and TNF-α production (Supplementary Fig. 3A–C), suggesting that Ori doesnot inhibit LPS-induced priming at the doses, which are effective for NLRP3 inhibition. Indeed, we found that the inhibitory activity of Ori on IL-1β release was about ten times more potent than its activity on TNF-α production (Supplementary Fig. 4A, B), suggesting that Ori has a robust inhibitory activity on NLRP3 inflammasome. Thus, these results indicate that Ori is a specific NLRP3 inflammasome inhibitor.

**Ori inhibits NLRP3 inflammasome assembly.** Next, we studied the mechanism of how Ori blocked NLRP3 activation. Firstly, we found that Ori could suppress nigericin-induced ASC oligomerization (Fig. 2a), an essential step for NLRP3 activation[36,37], suggesting that Ori acts upstream of ASC oligomerization to suppress NLRP3 activation. Secondly, Ori could not inhibit nigericin-induced mitochondrial damage and ROS production (Supplementary Fig. 5A), which are also the upstream signaling events of NLRP3 activation[16,38]. Moreover, we also found that Ori could not block nigericin-induced potassium or chloride efflux (Supplementary Fig. 5B, C), another two upstream signaling events of NLRP3 activation[39–41]. Thus, these results indicate that Ori doesnot affect the signaling events of upstream of NLRP3.

We then evaluated the effects of Ori on the formation NLRP3 inflammasome. One essential step for NLRP3 inflammasome assembly is the interaction between NEK7 and NLRP3, which is critical for the subsequent NLRP3 oligomerization and recruitment of ASC to NLRP3[42–44]. We then examined whether Ori could inhibit nigericin-induced endogenous NLRP3 inflammasome complex formation and found that it blocked both endogenous NEK7–NLRP3 and NLRP3-ASC interaction (Fig. 2b, c), suggesting that Ori might prevent NLRP3 inflammasome assembly by blocking NLRP3–NEK7 interaction. We further tested whether Ori could prevent the direct NEK7–NLRP3 interaction. Indeed, we found that Ori treatment inhibited NLRP3-NEK7 in HEK-293T cells (Fig. 2d). Moreover, the interaction byween purified NLRP3 and NEK7 could also be blocked by Ori (Fig. 2e and Supplementary Fig. 6A, B), suggesting that Ori directly prevents the NEK7–NLRP3 interaction. In contrast, Ori could not inhibit the direct NLRP3–NLRP3 interaction in HEK-293T cells (Fig. 2f), suggesting that Ori doesnot inhibit NLRP3 oligomerization. Consistent with this, Ori had no effects on ATPase activity of NLRP3 (Supplementary Fig. 6C), which is essential for NLRP3 oligomerization[45]. Moreover, Ori could not block NLRP3-ASC interaction in HEK-293T cells (Fig. 2g). NEK7 has been reported to interact with NEK9 during mitosis[46,47], but we found that Ori could not block the NEK7–NEK9 interaction (Supplementary Fig. 6D). Thus, these results demonstrate that Ori suppresses NLRP3 inflammasome activation by directly preventing the NEK7–NLRP3 interaction.

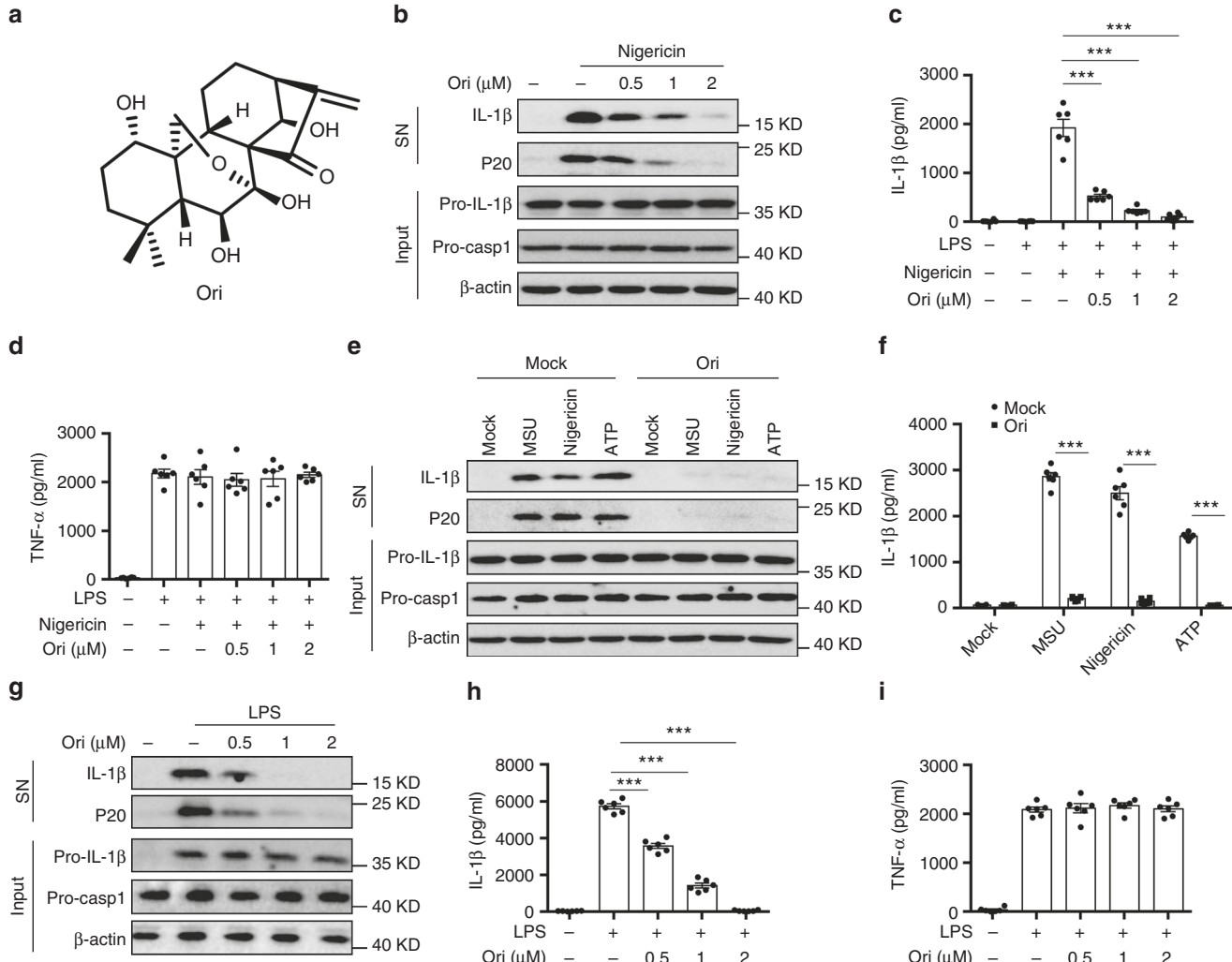

**Fig. 1** Ori blocks NLRP3 activation. **a** Ori structure. **b** BMDMs were primed with LPS and then stimulated with nigericin for 0.5 h with or without Ori. Western blot analysis of cleaved IL-1β, activated caspase-1 (P20) in culture supernatants (SN) and pro-IL-1β, pro-caspase-1 in lysates (Input) of BMDMs. **c**, **d** IL-1β (**c**) or TNF-α (**d**) production (as measured by ELISA) in SN from BMDMs primed with LPS and stimulated with nigericin for 0.5 h with or without Ori. **e**, **f** Western blot analysis (**e**) of cleaved IL-1β and activated caspase-1 or ELISA (**f**) of IL-1β in SN from BMDMs primed with LPS and stimulated with MSU, nigericin or ATP in the presence or absence of Ori (2 μM). **g**–**i** Western blot analysis of cleaved IL-1β and activated caspase-1 (**g**) or ELISA of IL-1β (**h**) or TNF−α (**i**) production in SN from PBMCs pretreated with Ori for 0.5 h and stimulated with LPS for 16 h. Data are expressed as mean and s.e.m (*n* = 6) from three independent experiments (**c**, **d**, **f**, **h**, **i**) or are representative of three independent experiments (**b**, **e**, **g**). Statistical differences were calculated by unpaired Student's *t*-test: ***P < 0.001

**Ori directly binds to NLRP3**. Since Ori prevents NLRP3 inflammasome assembly by preventing NLRP3–NEK7 interaction, it is possible that Ori directly targets NLRP3 or NEK7. When the cell lysates of LPS-primed BMDMs were incubated with a synthesized biotinylated Ori (bio-Ori), NLRP3, but not NEK7, ASC or caspase-1 was pulled down by bio-Ori (Fig. 3a and supplementary Fig. 7A). Moreover, the pull-down of NLRP3 by bio-Ori could be competed off by free Ori (Fig. 3b). To confirm the specificity of Ori–NLRP3 interaction, we incubated the lysates from LPS-primed BMDMs or HEK-293T cells overexpressing Flag-NLRP3 with bio-Ori and then detected by Strep-HRP. The data showed NLRP3 was the major protein, which could be bound by Ori (Fig. 3c, d). To check whether Ori directly binds to NLRP3, purified NLRP3 protein was incubated with bio-Ori and GFP-NLRP3 could be pulled down by bio-Ori (Fig. 3e), confirming that Ori directly interacts with NLRP3. In order to more precisely validate Ori as a direct NLRP3 inhibitor, we utilized

microscale thermophoresis (MST) assay to measure the direct interaction between Ori and purified GFP-NLRP3. The equilibrium dissociation constant ($K_D$) between Ori and purified GFP-NLRP3 was around 52.5 nM (Fig. 3f).

We also investigate the interaction between Ori and other inflammasome sensors and found that only NLRP3 could be pulled down by bio-Ori, while other sensors, such as AIM2, NLRC4, and NLRP1, could not (Fig. 3g). Further studies showed that the NACHT domain could interact with Ori, but the LRR or PYD domain could not (Fig. 3h). These results demonstrate that Ori directly targets NLRP3 by binding to its NACHT domain.

**Ori targets Cys279 of NLRP3 via covalent bond formation**. To investigate the nature of Ori interaction with NLRP3, we sought to determine the reversibility of Ori inhibition of NLRP3 inflammasome. LPS-primed BMDMs were incubated with Ori for

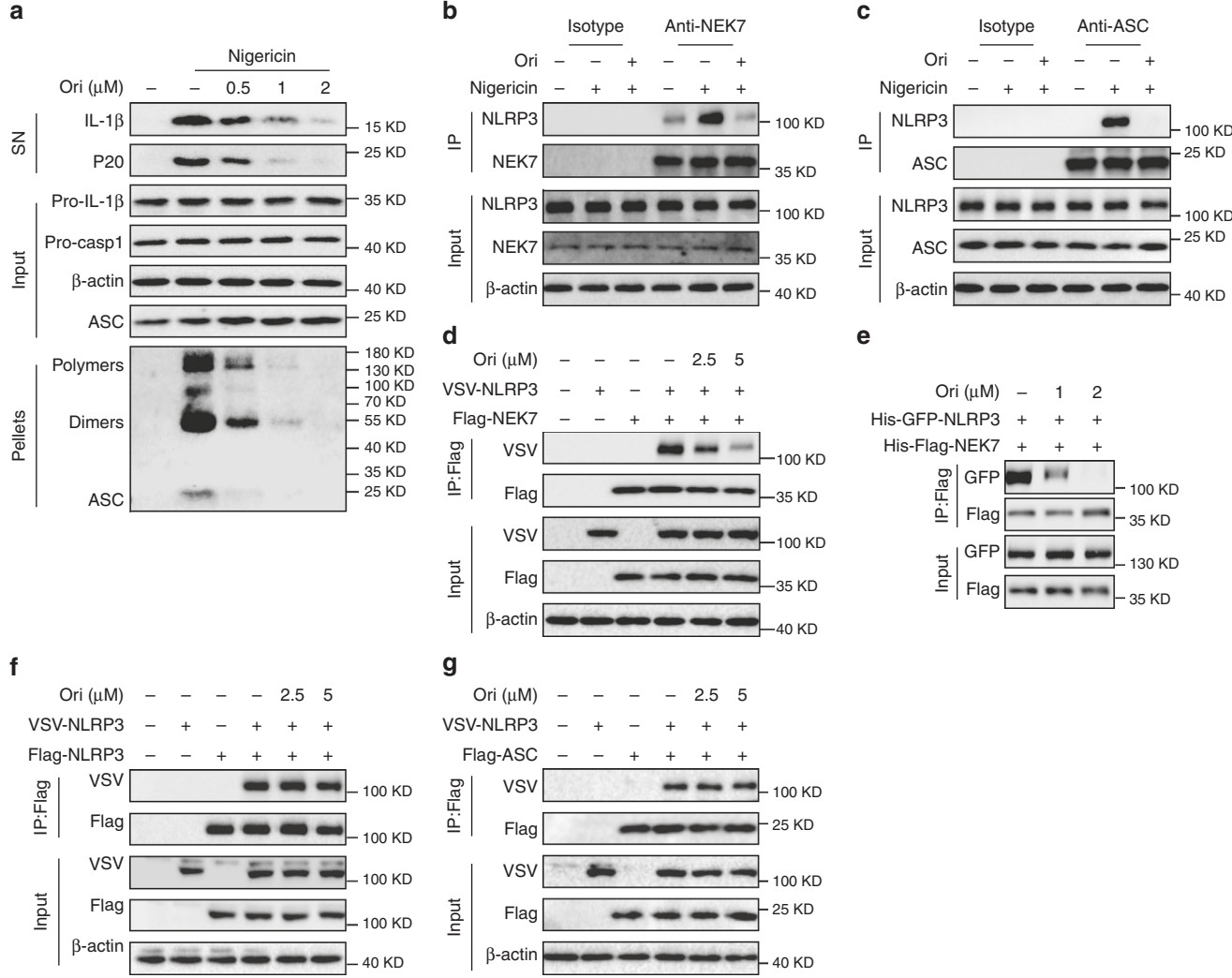

**Fig. 2** Ori inhibits NLRP3 inflammasome assembly by blocking NEK7–NLRP3 interaction. **a** Western blot analysis of cross-linked ASC in the Nonidet P-40–insoluble pellet of BMDMs stimulated with nigericin in the presence or absence of Ori. **b** An endogenous immunoprecipitation (IP) with NEK7 antibody or isotype antibody was performed in LPS-primed BMDMs stimulated with nigericin in the presence or absence of Ori. **c** An endogenous IP with ASC antibody or isotype antibody was performed in LPS-primed BMDMs stimulated with nigericin in the presence or absence of Ori. **d** IP and western blot analysis of NEK7–NLRP3 interaction in HEK-293T cells. **e** IP and western blot analysis of the interaction between purified NEK7 and NLRP3. **f** IP and western blot analysis of NLRP3 oligomerization in HEK-293T cells. **g** IP and western blot analysis of ASC-NLRP3 interaction in HEK-293T cells. Data are representative of three independent experiments

15 min and the performed three washes over 15 min to remove unbound drug before nigericin stimulation. The results showed that Ori still inhibited nigericin-induced IL-1β production after the washout (Fig. 4a), suggesting the inhibitory effects of Ori is irreversible. In contrast, the inhibitory effects of CY-09, another small-molecule inhibitor could directly bind to NLRP3[33], is reversible (Supplementary Fig. 8). We further tested whether Ori covalently bound to NLRP3. Western blot revealed that NLRP3 was pulled down by bio-Ori, which was reversed by adding an excess amount of free Ori. However, when lysates were pre-incubated with bio-Ori, post-treatment of an excess amount of Ori could not prevent NLRP3 binding to bio-Ori (Fig. 4b), indicating a covalent bond formation between Ori and NLRP3 protein. Because Ori contains an active carbon–carbon double-bond, which has a potential to react covalently with the thiols of cysteine on NLRP3.

Next, we used BLAST to analyze human NLRP3 NACHT protein sequence and found nine cysteine residues. To determine

which cysteine residue was responsible for the binding, we constructed NLRP3 mutants, in which a cysteine was changed to alanine. The results showed that mutation at cysteine 279 (C279A) abolished NLRP3 binding to Ori (Fig. 4c). In addition, although C279A mutant NLRP3 still bound to NEK7, Ori could not block this interaction (Fig. 4d). More importantly, we found that Ori could inhibit the activation of NLRP3 inflammasome in NLRP3-deficient macrophages, which had been reconstituted with WT NLRP3, but not mutant NLRP3 (C275A) (Fig. 4e, f). Thus, these results indicate that the cysteine 279 on NLRP3 is a covalent binding site of Ori.

Ori contains α,β-unsaturated carbonyl unit, which could serve as a Michael receptor for targeting the thiol of cysteine 279. To determine whether the carbon–carbon double-bond was responsible for the covalent bond formation between Ori and NLRP3, we synthesized the reduction form of Ori (R-Ori) where the carbon–carbon double-bond was reduced via hydrogenation (Fig. 5a). In contrast with Ori, R-Ori could not bind to NLRP3

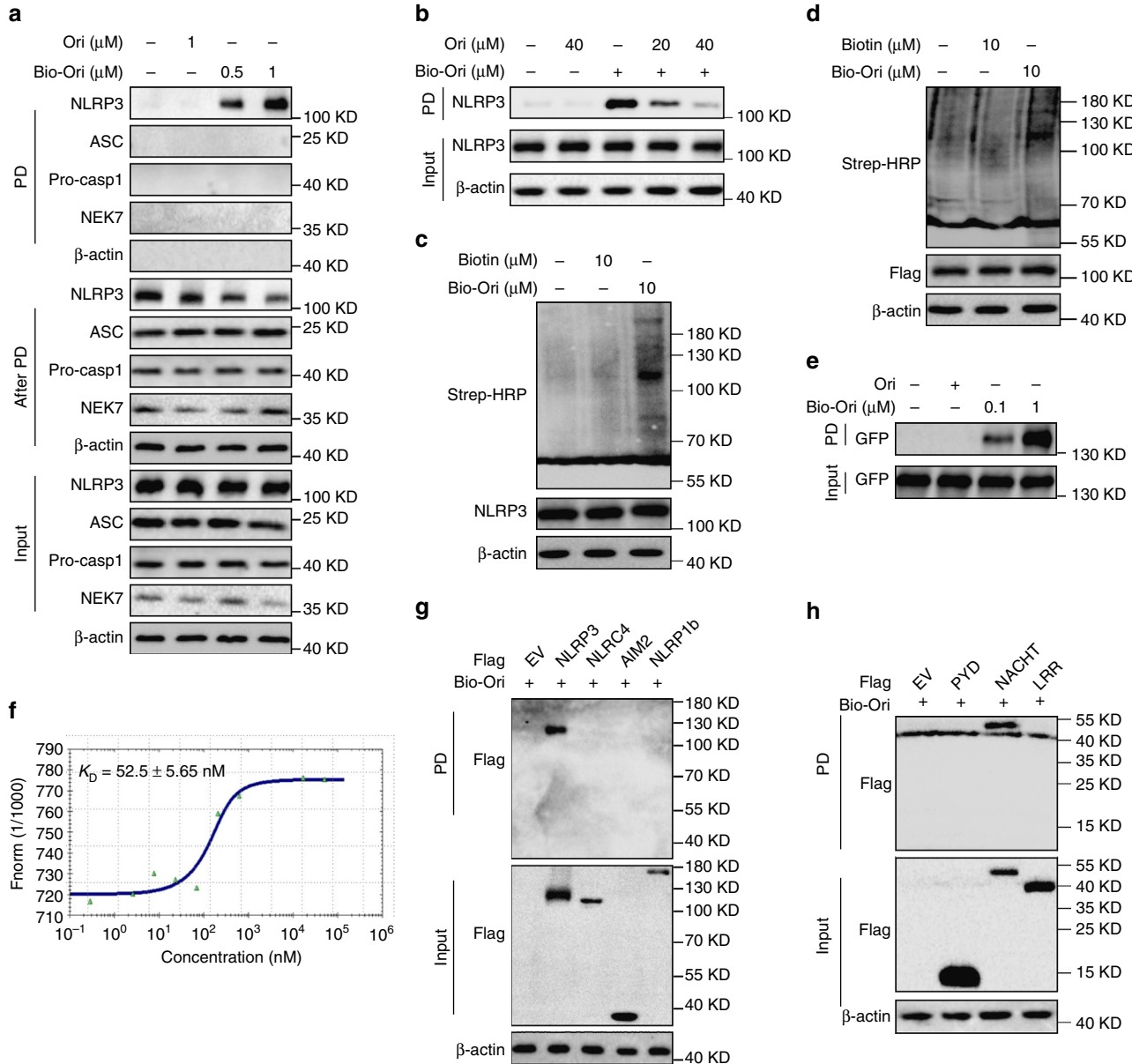

**Fig. 3** Ori binds to NLRP3 NACHT domain. **a** Cell lysates of LPS-primed BMDMs were incubated with bio-Ori for 2 h, which were then pulled down using streptavidin beads. The total proteins (input), bound proteins (PD) and remained proteins (After PD) were immunoblotted as indicated. **b** Cell lysates of LPS-primed BMDMs were incubated with bio-Ori (1 μM) and different concentrations of free Ori (20 μM, 40 μM), which were then pulled down using streptavidin beads. **c** Cell lysates of LPS-primed BMDMs incubated with bio-Ori or biotin and detected by Strep-HRP. **d** HEK-293T cell lysates overexpressing Flag-NLRP3 were incubated with bio-Ori or biotin and detected by Strep-HRP. **e** Purified human GFP-NLRP3 protein was incubated with indicated doses of bio-Ori (0.1 μM, 1 μM) and then pulled down using streptavidin beads. **f** MST assay for the affinity between Ori and purified GFP-NLRP3 protein. **g–h** Flag-tagged NLRP3, AIM2, NLRC4 or NLRP1 (**g**), NLRP3-LRR, NLPR3-NACHT or NLRP3-PYD (**h**) was expressed in HEK-293T cells. The HEK-293T cell lysates were incubated with bio-Ori (1 μM) and then were pulled down using streptavidin beads. Data are representative of two or three independent experiments

(Fig. 5b and Supplementary Fig. 7B). Consistent with this, reduction of Ori also abolished its inhibitory effects on NLRP3 inflammasome activation (Fig. 5c, d). Thus, these results indicate that the carbon–carbon double-bond is essential for the covalent bond formation between Ori and NLRP3.

**Ori suppresses NLRP3-dependent inflammation in vivo.** We next examined whether could inhibit NLRP3 inflammaosme activation in vivo. Previous study has shown that MSU can induce NLRP3-dependent IL-1β production and neutrophil influx in the abdominal cavity[24]. As expected, Ori treatment significantly alleviated MSU-induced IL-1β production and neutrophils migration in WT mice, but not in *Nlrp3*[-/-] mice (Fig. 6a–c). MSU deposition is the primary cause of gouty arthritis and delivery of MSU into the joints can result in NLRP3 inflammasome-dependent arthritis in mice[48,49]. Similarly, we found that Ori treatment prevented MSU injection induced acute joint swelling and IL-1β production in joint tissue of WT mice, but not in *Nlrp3*[-/-] mice (Fig. 6d, e). Thus, these results indicate

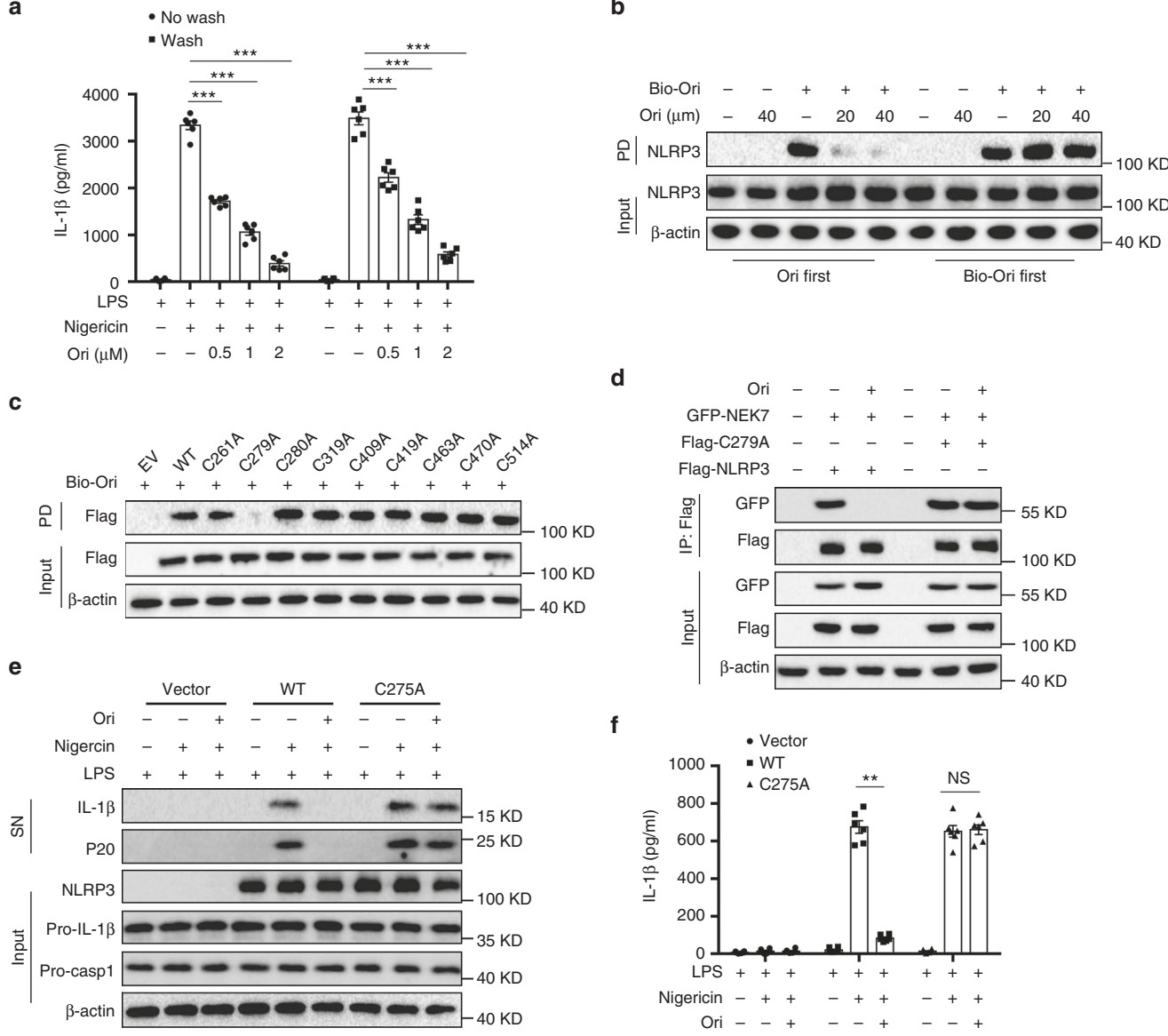

**Fig. 4** Ori covalently binds to the cysteine 279 on NLRP3. **a** ELISA of IL-1β in SN from LPS-primed BMDMs treated with different doses of Ori for 15 min and washed three times, then stimulated with nigericin. **b** Cell lysates of LPS-primed BMDMs were incubated with different concentrations of free Ori for 2 h before or after incubated with bio-Ori (1 μM) for 2 h, and then pulled down using streptavidin beads. **c** HEK-293T cells were transfected with Flag-tagged WT or indicated mutant NLRP3 constructs. The cell lysates were incubated with bio-Ori (1 μM) and then were pulled down using streptavidin beads. **d** IP and western blot analysis of the interaction between NEK7 and WT or mutant NLRP3 in HEK-293T cells. **e, f** Western blot analysis (**e**) of cleaved IL-1β and activated caspase-1 or ELISA of IL-1β (**f**) in SN from LPS-primed Nlrp3[-/-] BMDM reconstituted with WT or mutant NLRP3 (C275A) that were pretreated with Ori (2 μM) for 0.5 h and then stimulated with nigericin. Data are expressed as mean and s.e.m (n = 6) from three independent experiments (**a, f**) or are representative of three independent experiments (**b–e**). Statistical differences were calculated by unpaired Student's t-test: **P < 0.01, ***P < 0.001

that Ori prevents acute inflammation and tissue damage via inhibition of NLRP3 inflammasome.

NLRP3 inflammasome plays an important role in some chronic inflammation associated complex diseases, including T2D, Alzheimer's disease, and atherosclerosis[20–26,50]. We then tested the therapeutic effects of Ori in diabetic mice. WT mice and Nlrp3[-/-] mice were challenged with high-fat diet (HFD) for 12 weeks and then were given with Ori once a day at the dose of 3 mg/kg for 6 weeks. After 6-week, Ori-treated mice have shown less food intake and weight gain compared with the control group (Fig. 7a, b). The concentrations of fasting or basal blood glucose in diabetic WT mice were also reduced by Ori treatment (Fig. 7c, d). In addition, the insulin sensitivity in HFD-treated mice were

also improved by Ori (Fig. 7e, f). We also found that Ori reduced intracellular lipid accumulation in liver compared with the control mice (Supplementary Fig. 7). However, the therapeutic effects of Ori on metabolic disorders were absent in Nlrp3[-/-] mice (Fig. 7a–d, g, h and Supplementary Fig. 9). We also evaluated the effects of Ori on the metabolic parameters of healthy mice. C57BL/6J mice fed with normal diet were treated with Ori once a day at the dose of 3 mg/kg for 6 weeks and it had no effect on the metabolic parameters and serum chemistry (Supplementary Fig. 10). These results indicate that Ori can target NLRP3 to treat the metabolic disorders in diabetic mice.

In diabetic mice, NLRP3-dependent chronic inflammation contributes to T2D[51,52], we then examined whether Ori inhibited

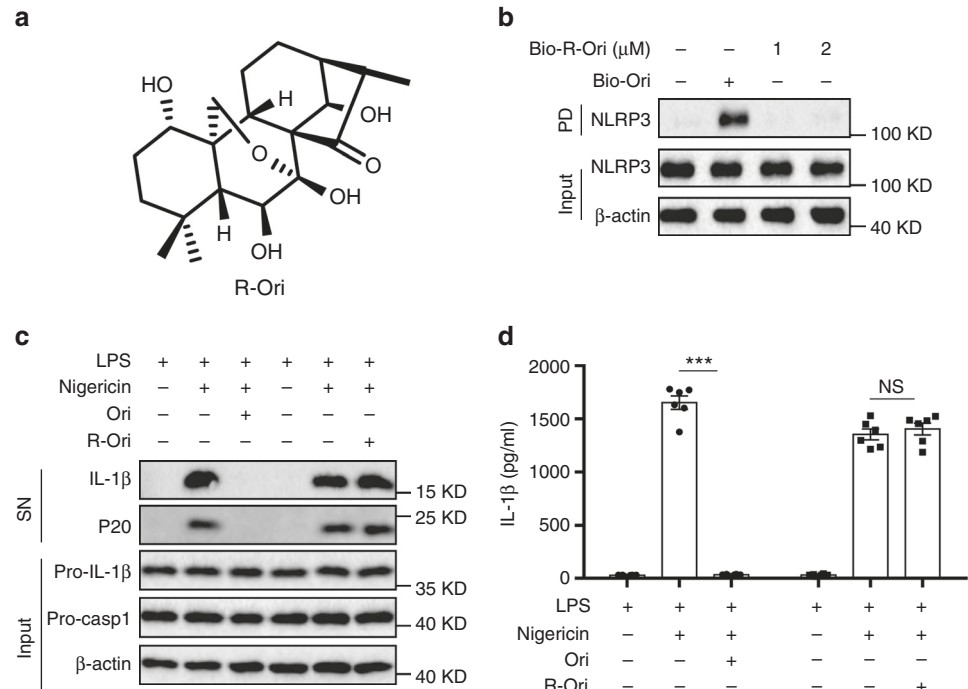

**Fig. 5** Ori binds to NLRP3 via its active carbon–carbon double-bond. **a** Structure of Ori without carbon–carbon double-bond (R-Ori). **b** Cell lysates of LPS-primed BMDMs were incubated with bio-Ori (1 μM) or indicated doses of bio-R-Ori for 2 h, and then pulled down using streptavidin beads. **c**, **d** Western blot analysis (**c**) of cleaved IL-1β and activated caspase-1 or ELISA of IL-1β (**d**) in SN from LPS-primed BMDMs pretreated with Ori (2 μM) or R-Ori (2 μM) for 0.5 h and then stimulated with nigericin. Data are expressed as mean and s.e.m ($n = 6$) from three independent experiments (**d**) or are representative of three independent experiments (**b**, **c**). Statistical differences were calculated by unpaired Student's $t$-test: ***$P < 0.001$

NLRP3-dependent chronic inflammation in diabetic mice. As expected, Ori inhibited HFD-induced IL-1β production in serum, liver or adipose tissues (Supplementary Fig. 11A–C). The HFD-induced caspase-1 activation observed in adipose tissue was also suppressed by Ori (Supplementary Fig. 11D). In addition, the inflammasome-independent cytokines, such as TNF-α and MCP-1, were also decreased by Ori treatment (Supplementary Fig. 11E–H), similar with results observed in $Nlrp3^{-/-}$ mice. These results suggest that Ori reduces NLRP3-dependent chronic inflammation in diabetic mice.

## Discussion

*Rabdosia rubescens* is a commonly used traditional Chinese medicine for treatment of inflammatory diseases[5,6], but the poorly known mechanism of action limits its clinical application. Here, we demonstrate that Ori, the major active constitute of *Rabdosia rubescens*, directly and covalently binds to NLRP3 and has remarkable anti-inflammasome activity both in vitro and in vivo, suggesting that Ori can be used as a lead to design new therapeutics against NLRP3-driven diseases.

Our results indicate that Ori targets NLRP3 to exert its anti-inflammatory activity. Although previous studies have shown that Ori can inhibit NF-κB or MAPK activation and suppress the production of inflammasome-independent cytokines, such as TNF-α and IL-6[7–9], our data showed that the doses needed for inhibition of TNF-α production were about ten times higher than the doses needed for NLRP3 inflammasome inhibition. Importantly, our results showed that the beneficial effects of Ori on peritonitis, gouty arthritis and type 2 diabetes were absent in $Nlrp3^{-/-}$ mice, suggesting that the in vivo anti-inflammatory activity of Ori depends on its inhibitory effects on NLRP3

inflammasome. These results suggest that *Rabdosia rubescens*, Ori or its derivatives might have a better therapeutic potential in NLRP3-driven inflammatory diseases than in NLRP3-independent inflammatory diseases.

Although both NLRP3 inflammasome components and upstream signaling events can be targeted, only targeting NLRP3 itself can specifically block its activation. Previous studies have identified several compounds, which can directly target NLRP3 itself, but all these compounds suppress NLRP3 activation by inhibition of its ATPase activity[33,45,53–56]. In contrast, our results showed that Ori could not inhibit the ATPase activity of NLRP3, although Ori directly bound NLRP3 inhibited NLRP3 inflammasome activation. In addition, the upstream signaling events of NLRP3 activation, such as potassium efflux and mitochondrial damage, were not affected by Ori treatment. Instead, we found that Ori could block the interaction between NLRP3 and NEK7, which is an essential step for NLRP3 inflammasome assembly[42–44]. Ori could not inhibit NLRC4 or AIM2 inflammasome activation, consistent with the previous reports showing that NEK7 interacted with NLRP3, but not with other inflammasome sensors, such as NLRC4 and AIM2[43]. Thus, our results indicate the interface between NLRP3 and NKE7 can be targeted to specifically inhibit NLRP3 inflammasome activation.

Our results demonstrate that Ori functions as a Michael receptor and covalently binds with NLRP3. We found that the α, β-unsaturated carbonyl unit of Ori was essential for its inhibitory effects on NLRP3 inflammasome activation, because selective reduction of the C=C double-bond abrogated its association with NLRP3 and also its inhibitory activity. Moreover, Ori could not bind with a mutant NLRP3, in which the cysteine 279 was substituted with alanine, suggesting that the covalent bond is formed

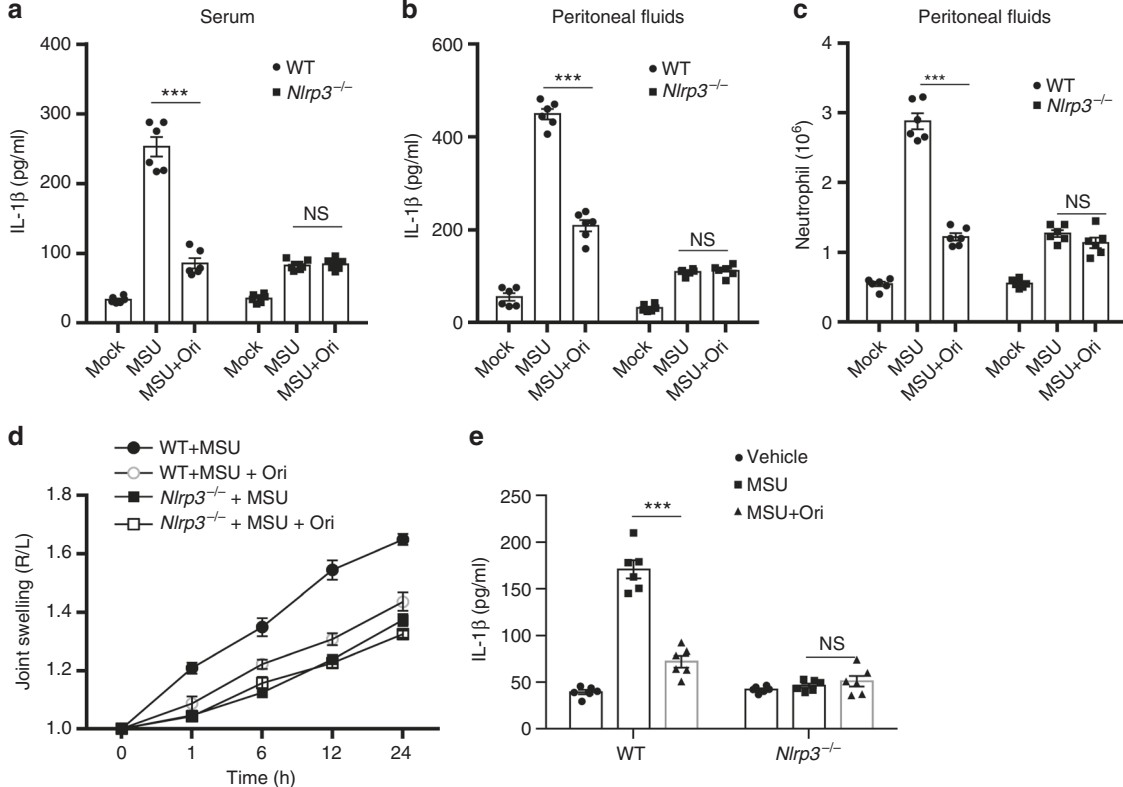

**Fig. 6** Ori suppresses NLRP3-dependent peritonitis and gouty arthritis. **a**, **b** ELISA of IL-1β in the serum (**a**) or peritoneal cavity (**b**) from WT or *Nlrp3*[−/−] mice intraperitoneally injected with MSU crystals (1 mg/mouse) in the presence or absence of Ori (20 mg/kg). Data expressed as mean and s.e.m ($n = 6$) and are representative of two independent experiments. **c** FACS analysis of neutrophil numbers in the peritoneal cavity from WT or *Nlrp3*[−/−] mice intraperitoneally injected with MSU crystals (1 mg/mouse) in the presence or absence of Ori (20 mg/kg). Data are expressed as mean and s.e.m ($n = 6$) and are representative of two independent experiments. **d**, **e** The joint swelling in different time points (**d**) or ELISA of IL-1β (**e**) in joint culture from WT or *Nlrp3*[−/−] mice, which have been intraarticularly injected with MSU with or without Ori (20 mg/kg). Data are expressed as mean and s.e.m ($n = 6$) and are representative of two independent experiments. Statistical differences were calculated by unpaired Student's *t*-test: **$P < 0.01$, ***$P < 0.001$

between the active double C=C and the cysteine 279 of NLRP3. Although caspase-1, which is the effector protease of NLRP3 inflammasome, contains a cysteine that is essential for its catalytic activity[57], it seems Ori cannot directly inhibit caspase-1, because it could not suppress AIM2 or NLRC4 inflammasome activation. The crystal structure of NLRP3 has not been reported, our efforts to clarify why Ori specifically binds with cysteine 279 of NLRP3 via co-crystal structure study were also not succeeded. Further studies need to be conducted.

Covalent drugs have been raised the safety concerns due to potential risk of idiosyncratic toxicity and/or immune-mediated drug hypersensitivity, but they have made a major impact on human health and many covalent drugs have been approved by US Food and Drug Administration (FDA). On the other hand, they can provide some pharmacological advantages, including enhanced potency and prolonged duration of action[58,59]. Thus, our results not only identify NLRP3 as the direct target of Ori, but also suggest that Ori or its analogs might have the potential to treat NLRP3-driven diseases, especially the chronic diseases.

## Methods
**Mice**. C57BL/6J mice were from the Model Animal Research Center of Nanjing University. *Nlrp3*[−/−] mice were described previously[24]. Mice were specific pathogen-free, maintained under a strict 12 h light cycle (lights on at 07:00 a.m. and off at 07:00 p.m.). All animal experiments protocols were approved by the Animal Care Committee of the University of Science and Technology of China.

**Reagents**. ATP, MSU, poly A:T, nigericin, and glucose were from Sigma. The human recombinant insulin was from Nova Nordisk. One Touch® Ultra® Blood Glucose Test System was from Roche. Oridonin (s2335) was bought from Selleck. MitoSOX, ultrapure LPS, Pam3CSK4 and MitoTracker were from Invitrogen. Protein G agarose and streptavidin-coated beads were, respectively, supplied by Millipore and Pierce Biochemicals. Anti-Flag (F2555) and anti-VSV (V4888) were bought from Sigma. Anti-NLRP3 (AG-20B-0014) and anti-mouse caspase-1 (p20) (AG-20B-0042) antibodies were from Adipogen. Anti-NEK7 (SC-50756) and anti-ASC (sc-22514-R) antibodies were obtained from Santa Cruz. Anti-β-actin (P30002) was bought from Abmart. Anti-human caspase-1 was from Cell Signaling Technology. Anti-human cleaved IL-1β (A5208206) was from Sangon Biotech. Anti-mouse IL-1β (p17) (AF-401-NA) was from R&D Systems. A standard high-fat diet (D12492, 60% kcal fat) was from Research Diet Company.

**Chemical synthesis**. For synthesis of R-Ori, Ori (30 mg) was dissolved in methanol (3 ml). After adding 10% Pd/C (3 mg), the mixture was stirred for 1 h under hydrogen atmosphere at room temperature. After filtering, the filtrate was concentrated under reduced pressure. The crude product was purified by flash chromatography, eluting with dichloromethane-methanol (v/v = 30:1), to give the desired product R-Ori (8.6 mg, 28.5%) as a white solid.

For synthesis of bio-Ori, Ori (18.2 mg, 0.05 mmol) was mixed with D-Biotin (13.4 mg, 0.055 mmol), 1-ethyl-3-(3-dimethylaminopropyl) carbodiimide (EDCI) (11.5 mg, 0.06 mmol) and 4-dimethylaminopyridine (DMAP) (5 mg) in 2 ml of dichloromethane. And the reaction was stirred at room temperature for 8 h. Then the resulting mixture was evaporated under reduced pressure. The residue was purified by pre-high performance liquid chromatography (HPLC) to afford bio-Ori (6.2 mg, trifluoroacetic acid (TFA) salt, 17.6%) as a white solid.

For synthesis of bio-R-Ori, R-Ori (36.6 mg, 0.1 mmol) was mixed with D-Biotin (26.8 mg, 0.11 mmol), EDCI (23 mg, 0.12 mmol) and DMAP (5 mg) in 2 ml of dichloromethane. And the reaction was stirred at room temperature for 3 h. Then the resulting mixture was evaporated under reduced pressure. The residue was purified by pre-HPLC to afford bio-R-Ori (28 mg, TFA salt, 39.7%) as a white solid.

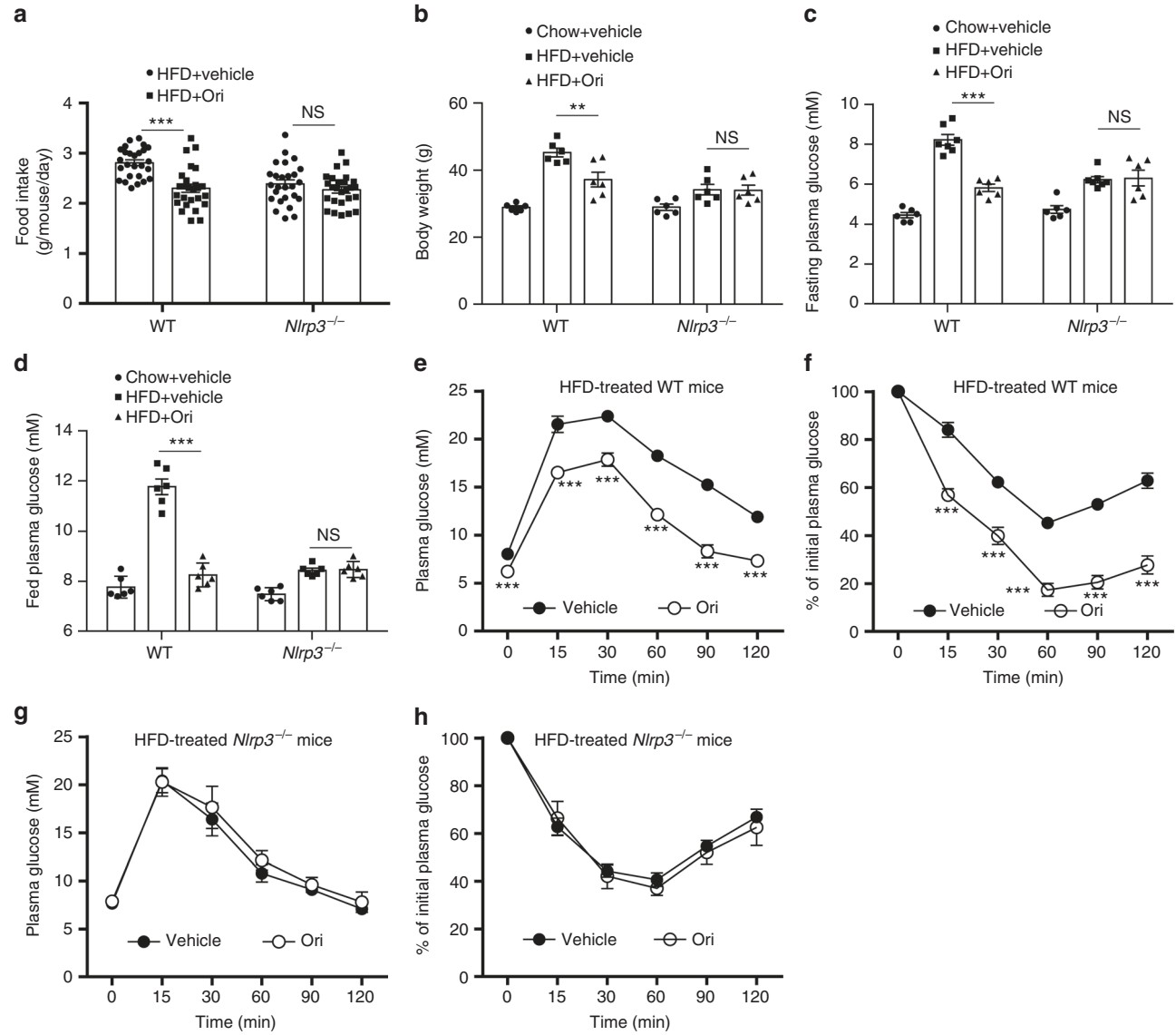

**Fig. 7** Treatment of metabolic disorders in HFD-induced diabetic mice with Ori. **a, b** Food intake and body weights of 16-week HFD-fed WT or *Nlrp3*^-/- mice after 6-week vehicle or Ori treatment. $n = 26$ per group. **c, d** Fasting (**c**) or fed (**d**) blood glucose levels of 16-week HFD-fed WT or *Nlrp3*^-/- mice treated with vehicle or Ori for 6 weeks. $n = 6$ per group. **e–h** GTT (**e, g**) or ITT (**f, h**) of 16-week HFD-fed WT or *Nlrp3*^-/- mice treated with vehicle or Ori for 6 weeks. Data are expressed as mean and s.e.m and are representative of two independent experiments. Statistical differences were calculated by unpaired Student's *t*-test: **$P < 0.01$, ***$P < 0.001$, NS, not significant

R-Ori:
$^1$H NMR (600 MHz, DMSO-$d_6$):δ (ppm) 6.74 (s, 1 H), 6.10 (d, $J = 2.3$ Hz, 1 H), 5.74 (d, $J = 11.0$ Hz, 1 H), 4.80 (s, 1 H), 4.35 (d, $J = 5.0$ Hz, 1 H), 4.07 (dd, $J = 10.1$, 1.4 Hz, 1 H), 3.83 (dd, $J = 10.2$, 1.8 Hz, 1 H), 3.43–3.39 (m, 1 H), 2.95–2.85 (m, 1 H), 2.38–2.32 (m, 1 H), 2.05–2.95 (m, 1 H), 1.92–1.84 (m, 1 H), 1.63–1.58 (m, 1 H), 1.54–1.41 (m, 4 H), 1.35–1.29 (m, 1 H), 1.25–1.23 (m, 2 H), 1.10–1.07 (m, 1 H), 1.02 (d, $J = 7.1$ Hz, 3 H), 1.00 (s, 3 H), 0.97 (s, 3 H).$^{13}$C NMR (150 MHz, DMSO-$d_6$) δ (ppm) 224.75, 96.64, 73.29, 73.14, 71.55, 62.84, 61.08, 60.26, 52.66, 44.84, 40.22, 38.45, 38.38, 33.42, 32.58, 29.35, 21.50, 19.15, 18.37, 10.22. MS (ESI) *m/z*: 367 [M + H]$^+$.HRMS (ESI) calcd. for C20H30O6, 367.2121; found, 367.2109.
Bio-Ori:
$^1$H NMR (600 MHz, DMSO-$d_6$):δ (ppm) 6.41 (s, 2 H), 5.99 (s, 1 H), 5.84 (s, 1 H), 4.33–4.26 (m, 2 H), 4.15–4.10 (m, 1 H), 4.09–4.03 (m, 1 H), 3.85–3.82 (m, 1 H), 3.52–3.48 (m, 1 H), 3.36–3.30 (m, 2 H), 3.10–3.04 (m, 1 H), 2.97–2.92 (m, 1 H), 2.85–2.77 (m, 1 H), 2.59–2.55 (m, 1 H), 2.30–2.19 (m, 1 H), 2.15–2.10 (m, 2 H), 1.85–1.80 (m, 1 H), 1.75–1.66 (m, 1 H), 1.55–1.41 (m, 8 H), 1.36–1.25 (m, 3 H), 1.23–1.16 (m, 1 H), 1.15–1.09 (m, 1 H), 1.03 (d, $J = 6.1$ Hz, 2 H), 1.00 (s, 3 H), 0.99 (s, 3 H).$^{13}$C NMR (150 MHz, DMSO-$d_6$) δ (ppm) 207.65, 172.04, 163.19, 151.31, 96.19, 95.87, 74.51, 73.64, 72.04, 62.96, 62.49, 62.26, 61.49, 59.68, 59.54, 55.78, 55.50, 54.32, 49.05, 41.86, 40.93, 38.82, 34.24, 33.78, 33.17, 28.42, 28.39,

25.92, 24.54, 22.11.MS (ESI) *m/z*: 591 [M + H]$^+$.HRMS (ESI) calcd. for C30H42N2O8S, 591.2740; found, 591.2728.
Bio-R-Ori:
$^1$H NMR (600 MHz, DMSO-$d_6$):δ (ppm) 6.39 (s, 2 H), 5.93 (d, $J = 1.7$ Hz, 1 H), 4.33–4.29 (m, 2 H), 4.17–4.12 (m, 1 H), 4.07–4.03 (m, 1 H), 3.83–3.79 (m, 1 H), 3.41 (d, $J = 5.9$ Hz, 1 H), 3.35–3.33 (m, 1 H), 3.15–3.05 (m, 1 H), 2.89–2.80 (m, 2 H), 2.58 (d, $J = 12.4$ Hz, 1 H), 2.43–2.38 (m, 1 H), 2.30–2.15 (m, 2 H), 2.15–2.05 (m, 1 H), 1.98–1.90 (m, 1 H), 1.75–1.67 (m, 1 H), 1.65–1.57 (m, 1 H), 1.57–1.40 (m, 7 H), 1.36–1.29 (m, 4 H), 1.23–1.17 (m, 1 H), 1.17–1.15 (m, 1 H), 1.12 (d, $J = 5.8$ Hz, 1 H), 1.03 (d, $J = 7.0$ Hz, 3 H), 1.00 (d, $J = 4.0$ Hz, 1 H), 0.98 (s, 3 H), 0.95 (s, 3 H).$^{13}$C NMR (150 MHz, DMSO-$d_6$):δ (ppm) 223.74, 171.61, 162.66, 95.40, 74.12, 73.85, 71.25, 62.68, 61.41, 60.99, 60.27, 59.18, 55.25, 55.02, 53.29, 45.80, 39.99, 38.31, 37.16, 33.87, 33.32, 32.32, 29.19, 27.96, 27.93, 24.06, 21.18, 19.24, 18.60, 9.86.MS (ESI) *m/z*: 593 [M + H]$^+$.HRMS (ESI) calcd. for C30H44N2O8S, 593.2897; found, 593.2886.

**Human samples**. The adult peripheral blood samples were obtained from four healthy donors at Hefei Blood Bank and the experimental protocols were performed according to the approved guidelines established by the Institutional

Human Research Subjects Protection Committee of the Ethics Committee of the University of Science and Technology of China.

**Cell preparation and stimulation**. BMDMs were isolated from 6–8 weeks-old mice bone marrow and cultured for 6 to 7 days in Dulbecco modified Eagle medium (DMEM) supplemented with 10% FBS and 30% SN from L929 cells[60]. DMEM supplemented with 10% FBS was used to culture HEK-293T (ATCC) and L929 (ATCC), which were not authenticated in our lab, but routinely tested for mycoplasma contamination. Human Lymphocyte Separation Medium (catalog no. P8610-200, Solarbio) was used to obtain human PBMCs. PBMCs were cultured overnight before stimulation in RPMI 1640 medium supplemented with 10% FBS and antibiotics.

To stimulate NLRP3 inflammasome, $5 \times 10^5$/ml BMDMs and $6 \times 10^6$/ml PBMCs were plated in 12-well plates. After 12–18 h, the cells were treated with 50 ng/ml LPS or 400 ng/ml Pam3CSK4 (for non-canonical inflammasome activation) for 3 h. After that, the cells were treated with Ori for 30 min and then were used for inflammasome stimulation as previously reported[61].

**Confocal microscopy**. In all, $2 \times 10^5$/ml BMDMs were plated on coverslips (Thermo Fisher Scientific) in 12-well plates overnight. After 12–18 h, the medium was changed to Opti-MEM with 1% FBS and stimulated with LPS (50 ng/ml) for 3 h. After that, Ori was added as described for another 0.5 h. BMDMs were stimulated by nigericin and stained with MitoTracker Red (50 nM) or Mitosox (5 μM), then cells were washed three times by ice-cold PBS and fixed with 4% PFA in PBS for 15 min. After that, cells were washed with PBST for three times. Confocal microscopy analysis were carried out by using a Zeiss LSM 700.

**Enzyme-linked immunosorbent assay (ELISA)**. The ELISA assay has been described previously[61].

**MSU-induced peritonitis and gouty arthritis**. In all, 6–8-weeks-old C57BL/6 mice were used to induce peritonitis by intraperitoneal administration of 1 mg MSU crystals (dissolved in 0.5 ml PBS). Before injection of MSU, 20 mg/kg Ori (dissolved in vehicle containing 90% PBS and 10% DMSO) were injected intraperitoneally. After 6 h, the mice were killed and 10 ml ice-cold PBS were used to wash the peritoneal cavities. The polymorphonuclear neutrophils in peritoneal lavage fluid was analyzed by flow cytometry by staining Ly6G and CD11b. The IL-1β level in serum or peritoneal lavage fluid was determined by ELISA.

For inducing joint inflammation, mice were administered Ori (20 mg/kg, dissolved in DMSO) by intraarticular injection. After 30 min, 0.5 mg MSU (dissolved in 20 μL PBS) was administrated intraarticularly and then the size of joints was measured at different time points. After 24 h, the patella were isolated and cultured in 200 μl opti-MEM medium containing 1% Penicillin-Streptomycin at room temperature for 1 h.

**Intracellular potassium or chloride determination**. For determination of intracellular potassium, BMDMs plated in 6-well plates were stimulated as normal. Culture medium was removed and cells were washed three times in potassium-free buffer (139 mM NaCl, 1.7 mM NaH2PO4, and 10 mM Na2HPO4, pH 7.2), and ultrapure HNO$_3$ was added to lyse the cells. Samples were transferred to glass bottles and then boiled for 30 min at 100 ℃. After that, the samples were add ddH2O to 5 ml. PerkinElmer Optima 2000 DV spectrometer was used to measure intracellular K$^+$. The protocol for the determination of intracellular chloride has been described previously[41].

**Plasmid constructions**. PCR reactions were performed using the PrimeSTAR® Max DNA Polymerase (TaKaRa). Restriction enzymes were purchased from Thermo Fisher Scientific. Gene recombination was performed using the ClonExpress® II One Step Cloning Kit (Vazyme). All fragments amplified by PCR were sequenced in the final plasmids. All primer sequences used are listed in Supplementary Table 1. To generate Flag-NEK9, the fragments of *NEK9* were amplified by PCR using primer pairs Flag-NEK9 1Forward/1Reverse using Human spleen genomic DNA as template. Flag-NEK9 2Forward/2Reverse were used to generate linear DNA of pCR3-Flag. Gene recombination was performed using the ClonExpress® II One Step Cloning Kit (Vazyme). To generate NLRP3 mutants, primers were used to generate linear DNA of pCR3-Flag-NLRP3, Gene recombination was performed using the ClonExpress® II One Step Cloning Kit (Vazyme).

To generate pLEX-NLRP3, *SpeI/XhoI* were used to digest pLEX vector (Thermo Fisher). linear DNA of *NLRP3* was generated by using Plex-NLRP3 1Forward/1Reverse. Gene recombination was performed using the ClonExpress® II One Step Cloning Kit (Vazyme). To generate pLEX-NLRP3(C275A), primers were used to generate linear DNA of pLEX-NLRP3. Gene recombination was performed using the ClonExpress® II One Step Cloning Kit (Vazyme).

**Protein expression and purification**. The protocol for expression and purification of His-GFP-NLRP3 and His-Flag-NEK7 has been described previously[33].

**Microscale thermophoresis assay**. The $K_D$ value was measured by using the Monolith NT.115 instrument (NanoTemper Technologies) as described previously[33].

**Immunoprecipitation and pull-down assay**. The protocols for immunoprecipitation and pull-down assay have been described previously[61] Uncropped immune blotting results are included in Supplementary Figs. 12–20.

**NLRP3 ATPase activity**. The assay for NLRP3 ATPase activity has been described previously[61].

**ASC oligomerization assay**. The assay for ASC oligomerization has been described previously[61].

**NLRP3 reconstitution**. *Nlrp3*$^{-/-}$ BMDMs were plated in 12-well plates for overnight, and then were lentivirally transduced with cDNA encoding mouse NLRP3 or NLRP3(C275A) using pLEX vector (Thermo Fisher). After 6 h, the culture medium was replaced by DMEM supplemented with 10% FBS and 30% supernatant from L929 cells. After 2 days, BMDMs were stimulated as normal.

**Detection of proteins bound by bio-Ori**. The cell lysates from BMDMs that have been stimulated with LPS for 3 h or HEK-293T cells, which have been transfected with Flag-NLRP3 plasmids for 24 h were collected and lysed and then centrifuged at 8000 rpm for 10 min. The supernatant was incubated with bio-Ori or Biotin for 2 h and then were analyzed by immunoblot and detected by Strep-HRP.

**High fat diet and Ori treatment**. Six-week-old WT or *Nlrp3*$^{-/-}$ mice with similar body weights and plasma glucose levels were randomized into different groups. After feeding with HFD for 12 weeks, the mice were treated with Ori at the dose of 3 mg/kg (dissolved in vehicle containing 90% PBS and 10% DMSO) by intraperitoneal injection once a day for 6 weeks. During Ori treatment and the subsequent experiments, the mice were maintained with HFD.

**Blood glucose assay**. The blood glucose assay has been described previously[61].

**Glucose tolerance test and insulin tolerance test**. The GTT and ITT assay have been described previously[61].

**Histological analysis**. The histological assay has been described previously[61].

**Statistical analyses**. The values are expressed as mean ± s.e.m. The unpaired *t*-test (GraphPad Software) were used for statistical analysis. The data points were not excluded. The researchers involved in this study were not blinded during sample collection or data analysis. Sample sizes were selected on the basis of preliminary results to ensure an adequate power. *P*-values < 0.05 were considered significant.

**Data availability**. The data that support this study are available within the article and its Supplementary Information files or available from the authors upon request.

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

## Acknowledgements

We thank Dr. Jurg Tschopp (University of Lausanne) for providing *Nlrp3*−/− mice. This work was supported by National Basic Research Program of China (2014CB910800), NSFC (81788101, 91742202, 81330078, 81525013, 81722022, 81571609), the Strategic Priority Research Program of the Chinese Academy of Sciences (XDPB03), the Young Talent Support Program and the Fundamental Research Funds for the Central Universities.

## Author contributions

H.H., H.J., Y.C., J.Y., and A.W. performed the experiments of this work. C.W., Q.L., G.L., W.J., X.D., and R.Z. designed the research. H.H., W.J., and R.Z. wrote the manuscript. R.Z. and W.J. supervised the project.

## Additional information

**Competing interests:** The authors declare no competing interests.

