## [Peer Review File · Nature Communications]

Reviewers' comments:

Reviewer #1 (Remarks to the Author):

Strategies to block the NLRP3 inflammasome are of interest to a number of subspecialties, as pathologic NLRP3 inflammasome activation is implicated not only in the pathogenesis of the autoinflammatory diseases but also in many common human diseases. The identification of safe inflammasome inhibitors is then a challenging question and is the focus of intense research: only a few candidates however have been reported to date. Thus, this study describing Oridonin (Ori) as a novel NLRP3 inflammasome blocking agent, is timely and potentially relevant. The comprehensive evaluation including the anti-inflammatory and metabolic effects of Ori in in vitro and in vivo models provide convincing data of a novel agent that may be beneficial in the treatment of some NLRP3 inflammasome mediated conditions. However, some points in the study are not convincing and should be investigated in more details.

1. A main weakness of the study is that most in vitro experiments have been done on murine cells, and PBMCs from a single healthy donor have been used, and only in the experiment shown in Figure 1 G-I. Since the variability of IL-1beta secretion among healthy individuals is high, more healthy donors should be tested. Moreover, in this experiment, PBMC were stimulated with LPS + Nigericin, a very artificial, non-physiologic way to induce inflammasome activation and IL-1beta secretion. Since human monocytes produce large amounts of IL-1beta after stimulation with LPS or other TLR agonists alone, without need of second signals, it would be important to test the effects of Ori in this more physiologic setting.

2. Figure 2D shows that 2µM of Ori completely prevents NLRP3-NEK7 interaction using purified proteins. However, in Figure 2D 5µM still shows some residual binding between the two proteins leaving open the possibility of secondary targets. How do the authors interpret such difference? It would be interesting to investigate the stability and the possible formation of active metabolites of the drug using cytochrome P450.

3. In Figure 2 the authors report that Ori prevents the interaction between NLRP3 and NEK7 (Figure 2D,E) but not NLRP3 and ASC (Figure 2F). As previously reported (Shi et al) and also mentioned by the authors in the main text, NEK7 binds to NLRP3, and this interaction is needed for the interaction of NLRP3 with ASC, oligomerization of ASC and activation of caspase-1. Thus, it is unclear why the inhibition of NEK7-NLRP3 interaction does not prevent also NLRP3-ASC interaction. How do the authors interpret these results?

3. Figure 7 shows that Ori treatment ameliorates the phenotype of HFD-induced metabolic syndrome in mice. A similar effect was also observed in the *nrlp3*^{-/-} mice. Interestingly, Figure 7A shows that Ori-treated mice and *nrlp3*^{-/-} mice have a lower amount of food intake throughout the duration of the study. How do the authors explain the reduced food intake in the Ori-treated mice and in the *nrlp3*^{-/-} mice? Is it possible that the benefits observed in the Ori-treated group following HFD is the reflection of the initial reduced intake of fatty acid, a well known inducer of the inflammasome? Have the authors measured baseline appetite hormones, i.e. leptin and ghrelin, in these mice to exclude genetic differences between the knock-out mice vs control?

Minor points

1. In the Introduction, the sentence about activation of NLRP3 inflammasome suggests that the various second signals act directly on NLRP3. The sentence should be modified to explain that the plethora of activating stimuli seems to converge on three main processes -intracellular ionic fluxes, ROS generation and lysosomal damage- but the precise mechanism of NLRP3 activation is still unclear

2. In Figure legend 2E the authors refer to VSV-NLRP3 but the figure shows GFP.

3. The inflammasome activation in BMDM should be described in more detail in the Methods section, particularly how the cells were cultured. It is not clear to this reviewer if the cells were cultured in presence or absence of FBS. The reference used to describe the methods does not include a method for this technique. It should be quoted the original manuscript where the technique was described.

4. Figure 2F and Figure 2G appear to be labeled incorrectly compared to the main text and the Figure legend.

5. Figure 3B and C legends should report the concentration used for Bio-Ori and Ori respectively. Figure 3C should report GFP-NLRP3.

Reviewer #2 (Remarks to the Author):

He et al provide a manuscript detailing studies of the ability of the diterpine, oridonin, to block NLRP3 based signaling. They demonstrate the compound specifically inhibits production of IL-1beta in response to NLRP3-activating stimuli but not stimuli that activate other inflammasomes capable of inducing IL-1beta secretion. They go on to demonstrate that oridonin directly and covalently binds NLRP3 and that this binding block interaction of NLRP3 with a NEK7 (a known cellular protein required for NLRP3 activation). Additionally the authors use two murine models of disease that depend on NLRP3 activation, The work is novel and important as inhibitors of the NLRP3 inflammasome may have great clinical utility in treating autoinflammatory disorders related to mutations in NLRP3 as well as for treating diseases caused by pathologic NLRP3 activation such as Gout. There are some concerns with the work as presented:

1) In the first sentence of the results section the authors indicate oridonin was identified as an NLRP3 inhibitor through a screen, however details of the screening assay and the results that led to identification of oridonin from the compound library are lacking. If these are not to be included in the manuscript, then the manuscript should simply lead with testing whether oridonin could inhibit NLRP3-mediated IL-1beta secretion.

2) The figure legend for figure 2 appears to be incorrect and the experiment performed and results for figure panels 2B and 2C are not understandable. The legend states that Ori was applied at various concentrations indicated above the gel. However, the concentration is not actually indicated on these panels. Further, the left three lanes of each panel show no immunoprecipitation of either protein blotted. It is unclear what IP antibodies were used in this experiment, this needs to be explained in the legend better. Even more importantly, the authors must explain why there is not precipitation of any protein in the left three lanes of each of the gels.

3) In order to get a better sense for the specificity of oridonin as a specific covalent inhibitor of NLRP3, it would be helpful to see immunoblots of monocyte lysates containing native NLRP3 and possibly cell line lysate overexpressing NLRP3 that were incubated with biotinylated oridonin using a labeled streptavidin as a detection agent to show that other proteins in lysate are not being covalently modified by oridonin. Additionally, in the pull down experiments in figure 3, it would be helpful to see the supernatants immunoblotted for NLRP3 after pull down in addition to the initial input to see the fraction of NLRP3 that is actually modified by oridonin treatment in these lysates

4) In describing the covalent binding of oridonin, the authors find that adding reducing agents (BME or DTT) prevents oridonin binding to NLRP3 (figure 4c), The authors need to explain why they believe these reducing agents block ori inhibition. It seems that under normal conditions, the cysteins in NLRP3 will be reduced in this intracellular protein. Are these agents acting on oridonin rather than a cysteine in NLRP3?

5) Although the authors present clear data demonstrating that cysteine 279 is the only cysteine in NLRP3 NACHT Domain that when mutated the protein fails to bind oridonin, it is unclear that this mutation is not simply disrupting the function of NLRP3, the mutation could cause alteration in NLRP3 structure or folding stability that leads to a molecule. The best experiment to demonstrate

this would be to express the mutant protein in NLRP3 deficient macrophages and show that NLRP3-dependent IL-1 β secretion is restored and that that secretion is now no longer sensitive to oridonin. In the absence of experimental support, the possibility that the disruption of oridonin binding due to this mutation could be because of an effect on protein structure/stability should at least be addressed in the discussion of the results as a limitation of the work presented.

6) There are very few details about the ITC experiments done to demonstrate direct binding of oridonin to NLRP3. The binding of a covalent (irreversible) inhibitor to other enzymes have been noted to release substantially more heat than non-covalent inhibitors (shown for the enzyme MAO-B by Rojas et al PMID: 25600407) Additional details on ITC data, including a comparison of this irreversible inhibitor to the authors previously identified reversible inhibitor of NLRP3 (PMID 29021150) would strengthen the argument that oridonin acts as a novel covalent modifier of NLRP3

Minor issues:

- 1) there are numerous English language based edits that the manuscript could benefit from a careful review prior to publication.
- 2) Reference lists contains a reference #59 that I do not actually seereferenced in the text (this may be a hold over from a prior version) if this is the case it should be removed
- 3) Figure 3 panel E refers to NLRC4 as its older original name: IPAF-this should be corrected to reference current nomenclature: NLRC4

Reviewers' comments:

Reviewer #1 (Remarks to the Author):

Strategies to block the NLRP3 inflammasome are of interest to a number of subspecialties, as pathologic NLRP3 inflammasome activation is implicated not only in the pathogenesis of the autoinflammatory diseases but also in many common human diseases. The identification of safe inflammasome inhibitors is then a challenging question and is the focus of intense research: only a few candidates however have been reported to date. Thus, this study describing Oridonin (Ori) as a novel NLRP3 inflammasome blocking agent, is timely and potentially relevant. The comprehensive evaluation including the anti-inflammatory and metabolic effects of Ori in in vitro and in vivo models provide convincing data of a novel agent that may be beneficial in the treatment of some NLRP3 inflammasome mediated conditions.

However, some points in the study are not convincing and should be investigated in more details.

1. A main weakness of the study is that most in vitro experiments have been done on murine cells, and PBMCs from a single healthy donor have been used, and only in the experiment shown in Figure 1 G-I. Since the variability of IL-1beta secretion among healthy individuals is high, more healthy donors should be tested. Moreover, in this experiment, PBMC were stimulated with LPS + Nigericin, a very artificial, non-physiologic way to induce inflammasome activation and IL-1beta secretion. Since human monocytes produce large amounts of IL-1beta after stimulation with LPS or other TLR agonists alone, without need of second signals, it would be important to test the effects of Ori in this more physiologic setting.

Reply: Thanks very much for the suggestion. In the revised manuscript, we have shown that Ori could not block LPS-induced NLRP3 inflammasome activation in PBMCs from four healthy donors (Fig. 1G-I and Supplementary Fig. 1).

2. Figure 2D shows that 2µM of Ori completely prevents NLRP3-NEK7 interaction using purified proteins. However, in Figure 2D 5µM still shows some residual binding between the two proteins leaving open the possibility of secondary targets. How do the authors interpret such difference? It would be interesting to investigate the stability and the possible formation of active metabolites of the drug using cytochrome P450.

Reply: Thanks very much for the suggestion.

1) In Fig. 2D, Ori were added to cell cultures to block NLRP3-NEK7 interaction, while in Fig. 2E, it was directly added to the buffer containing purified proteins. It is reasonable that higher doses of Ori were needed to suppress NLRP3-NEK7 interaction in cell cultures, because only a portion of Ori can get into the cells.

2) Regarding the pharmacokinetic property of Ori, it has been reported¹⁻³, so we don't repeat the experiments.

3. In Figure 2 the authors report that Ori prevents the interaction between NLRP3 and NEK7 (Figure 2D,E) but not NLRP3 and ASC (Figure 2F). As previously reported (Shi et al) and also

mentioned by the authors in the main text, NEK7 binds to NLRP3, and this interaction is needed for the interaction of NLRP3 with ASC, oligomerization of ASC and activation of caspase-1. Thus, it is unclear why the inhibition of NEK7-NLRP3 interaction does not prevent also NLRP3-ASC interaction. How do the authors interpret these results?

Reply: Thanks very much for the comments. In Fig.2G, NLRP3 and ASC were overexpressed in HEK-293T cells. In this system, the protein levels of NLRP3 or ASC are very high, so the interaction between them doesn't rely on the upstream signaling events. We used this system to test whether Ori could prevent the direct interaction between NLRP3-ASC.

In macrophages, NLRP3 agonists-induced NLRP3-ASC interaction depends on the upstream signaling events, such as NLRP3-NEK7 interaction. Indeed, we found that Ori treatment could inhibit nigericin-induced endogenous NLRP3-ASC interaction (Fig. 2C).

4. Figure 7 shows that Ori treatment ameliorates the phenotype of HFD-induced metabolic syndrome in mice. A similar effect was also observed in the *nrlp3*^{-/-} mice. Interestingly, Figure 7A shows that Ori-treated mice and *nrlp3*^{-/-} mice have a lower amount of food intake throughout the duration of the study. How do the authors explain the reduced food intake in the Ori-treated mice and in the *nrlp3*^{-/-} mice? Is it possible that the benefits observed in the Ori-treated group following HFD is the reflection of the initial reduced intake of fatty acid, a well known inducer of the inflammasome? Have the authors measured baseline appetite hormones, i.e. leptin and ghrelin, in these mice to exclude genetic differences between the knock-out mice vs control?

Reply: Thanks very much for the comments. The food intake data shown in Fig.7A was from the mice which have been treated with HFD for 3 months followed by 6-week Ori treatment. The reduced food intake in the Ori-treated mice or NLRP3 KO mice might be due to the improved metabolic status. Indeed, we have also compared the food intake between WT and NLRP3 KO mice during the first week of HFD treatment and found no differences (see the Fig.A below). Similarly, in the first-week treatment, Ori also had little effects on food intake in HFD-treated mice (see the Fig.B below). These results suggest that the reduced food intake in NLRP3 KO or Ori-treated mice is not due to the initial reduced intake of fatty acid or the genetic differences between WT or NLRP3 KO mice.

Minor points

1. In the Introduction, the sentence about activation of NLRP3 inflammasome suggests that the

various second signals act directly on NLRP3. The sentence should be modified to explain that the plethora of activating stimuli seems to converge on three main processes -intracellular ionic fluxes, ROS generation and lysosomal damage- but the precise mechanism of NLRP3 activation is still unclear

Reply: Thanks very much for the suggestion. We revised the sentence as " NLRP3 inflammasome can be activated by not only pathogen-associated molecular patterns (PAMPs), but also host-derived "danger signals", including monosodium urate crystals (MSU), cholesterol crystals, amyloid- β aggregates, unsaturated fatty acids, high glucose and ceramide. These activators seem to converge on three main processes intracellular ionic fluxes, ROS generation and lysosomal damage to activate NLRP3 inflammasome, but the precise mechanism of NLRP3 activation is still unclear. " in the revised manuscript.

2. In Figure legend 2E the authors refer to VSV-NLRP3 but the figure shows GFP.

Reply: We apologized for this error. We corrected this in the revised manuscript.

3. The inflammasome activation in BMDM should be described in more detail in the Methods section, particularly how the cells were cultured. It is not clear to this reviewer if the cells were cultured in presence or absence of FBS. The reference used to describe the methods does not include a method for this technique. It should be quoted the original manuscript where the technique was described.

Reply: Thanks very much for the suggestion. We have modified the Methods section and cited the original reference in the revised manuscript.

4. Figure 2F and Figure 2G appear to be labeled incorrectly compared to the main text and the Figure legend.

Reply: We apologized for this error. We corrected this in the revised manuscript.

5. Figure 3B and C legends should report the concentration used for Bio-Ori and Ori respectively. Figure 3C should report GFP-NLRP3.

Reply: Thanks very much for the suggestion. We revised these issues in the revised manuscript.

Reviewer #2 (Remarks to the Author):

He et al provide a manuscript detailing studies of the ability of the diterpine, oridonin, to block NLRP3 based signaling. They demonstrate the compound specifically inhibits production of IL-1 β in response to NLRP3-activating stimuli but not stimuli that activate other inflammasomes capable of inducing IL-1 β secretion. They go on to demonstrate that oridonin directly and covalently binds NLRP3 and that this binding block interaction of NLRP3 with a NEK7 (a known cellular protein required for NLRP3 activation. Additionally the authors use two murine models of disease that depend on NLRP3 activation, The work is novel and important as inhibitors of the NLRP3 inflammasome may have great clinical utility in treating autoinflammatory disorders related to mutations in NLRP3 as well as for treating diseases caused by pathologic NLRP3 activation such as Gout. There are some concerns with the work as

presented:

1) In the first sentence of the results section the authors indicate oridonin was identified as an NLRP3 inhibitor through a screen, however details of the screening assay and the results that led to identification of oridonin from the compound library are lacking. If these are not to be included in the manuscript, then the manuscript should simply lead with testing whether oridonin could inhibit NLRP3-mediated IL-1 β secretion.

Reply: Thanks for the comments. We revised the sentence as " To test whether oridonin (Ori) could block NLRP3 inflammasome activation, we first examined the effect of Ori on caspase-1 cleavage and IL-1 β secretion (Fig. 1A)" in the revised manuscript.

2) The figure legend for figure 2 appears to be incorrect and the experiment performed and results for figure panels 2B and 2C are not understandable. The legend states that Ori was applied at various concentrations indicated above the gel. However, the concentration is not actually indicated on these panels. Further, the left three lanes of each panel show no immunoprecipitation of either protein blotted . It is unclear what IP antibodies were used in this experiment, this needs to be explained in the legend better. Even more importantly, the authors must explain why there is not precipitation of any protein in the left three lanes of each of the gels.

Reply: We apologized for the missing labels in the figures. Figure 2B and 2C were endogenous IP in BMDMs treated with oridonin (2 μ M) and nigericin, IP antibodies were NEK7 (Figure 2B) and ASC (Figure 2C), IgG was used as isotype (the left three lanes). We modified the figures in the revised manuscript.

3) In order to get a better sense for the specificity of oridonin as a specific covalent inhibitor of NLRP3, it would be helpful to see immunoblots of monocyte lysates containing native NLRP3 and possibly cell line lysate overexpressing NLRP3 that were incubated with biotinylated oridonin using a labeled streptavidin as a detection agent to show that other proteins in lysate are not being covalently modified by oridonin. Additionally, in the pull down experiments in figure 3, it would be helpful to see the supernatants immunoblotted for NLRP3 after pull down in addition to the initial input to see the fraction of NLRP3 that is actually modified by oridonin treatment in these lysates

Reply: Thanks very much for the suggestions.

1) As suggested, the lysates of LPS-treated BMDMs or NLRP3-overexpressing HEK-293T cells were incubated with bio-Ori and then were detected by HRP- streptavidin. These results showed that NLRP3 is the major protein covalently modified by oridonin (Fig. 3C, D).

2) We examined NLRP3 protein in lysate after pull down, NLRP3 decreased but other proteins not (Fig.3A). These results mean that NLRP3 is actually modified by oridonin treatment in these lysates.

4) In describing the covalent binding of oridonin, the authors find that adding reducing agents (BME or DTT) prevents oridonin binding to NLRP3 (figure 4c), The authors need to explain why they believe these reducing agents block ori inhibition. It seems that under normal conditions, the cysteines in NLRP3 will be reduced in this intracellular protein. Are these agents acting on oridonin rather than a cysteine in NLRP3?

Reply: Thanks very much for the comments. We agree with the reviewer that these agents possibly also can acting on the cysteine in the NLRP3, so we removed these data in the revised manuscript.

5) Although the authors present clear data demonstrating that cysteine 279 is the only cysteine in NLRP3 NACHT Domain that when mutated the protein fails to bind oridonin, it is unclear that this mutation is not simply disrupting the function of NLRP3, the mutation could cause alteration in NLRP3 structure or folding stability that leads to a molecule. The best experiment to demonstrate this would be to express the mutant protein in NLRP3 deficient macrophages and show that NLRP3-dependent IL-1 β secretion is restored and that that secretion is now no longer sensitive to oridonin. In the absence of experimental support, the possibility that the disruption of oridonin binding due to this mutation could be because of an effect on protein structure/stability should at least be addressed in the discussion of the results as a limitation of the work presented.

Reply: Thanks very much for the suggestion. In the revised manuscripts, we provided new data (Figure 4F and 4G) showing that Ori could inhibit activation of NLRP3 inflammasome in NLRP3 deficient macrophages reconstituted with WT NLRP3, but not mutant NLRP3 (C275A).

6) There are very few details about the ITC experiments done to demonstrate direct binding of oridonin to NLRP3. The binding of a covalent (irreversible) inhibitor to other enzymes have been noted to release substantially more heat than non-covalent inhibitors (shown for the enzyme MAO-B by Rojas et al PMID: 25600407) Additional details on ITC data, including a comparison of this irreversible inhibitor to the authors previously identified reversible inhibitor of NLRP3 (PMID 29021150) would strengthen the argument that oridonin acts as a novel covalent modifier of NLRP3

Reply: Thanks very much for the comments.

The method we used in our manuscript is MicroScale Thermophoresis (MST). MST is based on thermophoresis, the directed movement of molecules in a temperature gradient, MST can detect molecular interactions at nM concentrations, even at pM concentrations⁴⁻⁶. ITC typically requires a protein concentration of 10 to 1000 μ M. Higher protein concentration ie 100 μ M produces a better signal in heat exchange. Considering the MW of full length NLRP3, which is over 110 kDa, the minimal sample requirement for a ITC-200 cell will be around 300 μ g NLRP3 for one test. In contrast, only about 3 μ g NLRP3 protein was needed for MST. Since it's difficult to obtain enough NLRP3 protein for ITC, we chose MST as the assay. However, MST cannot distinguish between reversible and irreversible binding.

To provide a control, we used CY-09, which is another NLRP3 inhibitor, as a control in washout experiments and found that CY-09 could not inhibit nigericin-induced IL-1 β release after the washout (supplementary Fig. 8).

Minor issues:

1) there are numerous English language based edits that the manuscript could benefit from a careful review prior to publication.

Reply: Thanks very much, we have modified the text in the revised manuscript.

2) Reference lists contains a reference #59 that I do not actually see referenced in the text (this may be a hold over from a prior version) if this is the case it should be removed.

Reply: Thanks very much, we have removed this reference in the revised manuscript.

3) Figure 3 panel E refers to NLRC4 as its older original name: IPAF-this should be corrected to reference current nomenclature: NLRC4

Reply: Thanks very much, we have corrected it in the revised manuscript.

Reference:

- 1 Xu, W. *et al.* Pharmacokinetic behaviors and oral bioavailability of oridonin in rat plasma. *Acta Pharmacol Sin* **27**, 1642-1646, doi:10.1111/j.1745-7254.2006.00440.x (2006).
- 2 Tian, T. *et al.* Identification of metabolites of oridonin in rats with a single run on UPLC-Triple-TOF-MS/MS system based on multiple mass defect filter data acquisition and multiple data processing techniques. *J Chromatogr B Analyt Technol Biomed Life Sci* **1006**, 80-92, doi:10.1016/j.jchromb.2015.10.006 (2015).
- 3 Gao, L. *et al.* Studies on pharmacokinetics and tissue distribution of oridonin nanosuspensions. *Int J Pharm* **355**, 321-327, doi:10.1016/j.ijpharm.2007.12.016 (2008).
- 4 Seidel, S. A. *et al.* Microscale thermophoresis quantifies biomolecular interactions under previously challenging conditions. *Methods* **59**, 301-315, doi:10.1016/j.ymeth.2012.12.005 (2013).
- 5 Jerabek-Willemsen, M., Wienken, C. J., Braun, D., Baaske, P. & Duhr, S. Molecular interaction studies using microscale thermophoresis. *Assay Drug Dev Technol* **9**, 342-353, doi:10.1089/adt.2011.0380 (2011).
- 6 Wienken, C. J., Baaske, P., Rothbauer, U., Braun, D. & Duhr, S. Protein-binding assays in biological liquids using microscale thermophoresis. *Nat Commun* **1**, 100, doi:10.1038/ncomms1093 (2010).

REVIEWERS' COMMENTS:

Reviewer #1 (Remarks to the Author):

The authors have addressed my queries; the ms is much improved and the data and conclusions are compelling

Reviewer #2 (Remarks to the Author):

The authors have addressed all the major concerns I raised with the initial manuscript. I believe the current submitted manuscript presents sufficient data supporting the conclusions presented by the authors regarding the finding that Oridonin is an inhibitor of NLRP3 activation that acts through a mechanism of covalent modification of NLRP3.

REVIEWERS' COMMENTS:

Reviewer #1 (Remarks to the Author):

The authors have addressed my queries; the ms is much improved and the data and conclusions are compelling

Reply: Thank you very much.

Reviewer #2 (Remarks to the Author):

The authors have addressed all the major concerns I raised with the initial manuscript. I believe the current submitted manuscript presents sufficient data supporting the conclusions presented by the authors regarding the finding that Oridonin is an inhibitor of NLRP3 activation that acts through a mechanism of covalent modification of NLRP3.

Reply: Thanks a lot.